# siRNA-Mediated MELK Knockdown Induces Accelerated Wound Healing with Increased Collagen Deposition

**DOI:** 10.3390/ijms24021326

**Published:** 2023-01-10

**Authors:** Lukasz Szymanski, Sławomir Lewicki, Tomasz Markiewicz, Szczepan Cierniak, Jean-Pierre Tassan, Jacek Z. Kubiak

**Affiliations:** 1Department of Molecular Biology, Institute of Genetics and Animal Biotechnology, Polish Academy of Sciences, 05-552 Magdalenka, Poland; 2Institute of Outcomes Research, Maria Sklodowska-Curie Medical Academy, 03-411 Warsaw, Poland; 3Faculty of Medical Sciences and Health Sciences, Kazimierz Pulaski University of Technology and Humanities, 26-600 Radom, Poland; 4Department of Patomorphology, Military Institute of Medicine, 01-755 Warsaw, Poland; 5Faculty of Electrical Engineering, Warsaw University of Technology, 00-662 Warsaw, Poland; 6UMR 6290, CNRS, Institute of Genetics and Development of Rennes, Dynamics and Mechanics of Epithelia Group, Faculty of Medicine, University Rennes, CEDEX, 35043 Rennes, France; 7Laboratory of Molecular Oncology and Innovative Therapies, Department of Oncology, Military Institute of Medicine, 01-755 Warsaw, Poland

**Keywords:** MELK, keratinocytes, fibroblasts, siRNA, mice, rodents, wound healing, proliferation

## Abstract

Skin wounds remain a significant problem for the healthcare system, affecting the clinical outcome, patients’ quality of life, and financial costs. Reduced wound healing times would improve clinical, economic, and social aspects for both patients and the healthcare system. Skin wound healing has been studied for years, but effective therapy that leads to accelerated wound healing remains to be discovered. This study aimed to evaluate the potential of MELK silencing to accelerate wound healing. A vectorless, transient knockdown of the MELK gene using siRNA was performed in a murine skin wound model. The wound size, total collagen, type 3 collagen, vessel size, vessel number, cell proliferation, cell apoptosis, number of mast cells, and immune infiltration by CD45, CD11b, CD45, and CD8a cells were evaluated. We observed that treatment with MELK siRNA leads to significantly faster wound closing associated with increased collagen deposition.

## 1. Introduction

Wound healing is a highly dynamic and well-coordinated process to restore the physical integrity of tissue after injury or infection. The process of wound healing can be divided into four major phases. Firstly, hemostasis is achieved by the activation of intrinsic and extrinsic coagulation pathways. The second phase, known as the inflammatory phase, starts even before the coagulation process is completed. It involves the influx of white blood cells and additional thrombocytes. Then, the third proliferative phase begins, during which neovascularization and re-epithelialization are observed. Finally, the remodeling phase begins, which leads to the maturation and restoration of the tissue’s strength [1,2,3]. The processes are complex and involve not only the intracellular machinery of individual cells but also appropriate interactions between various types of cells. Disturbances in such interactions lead to abnormal wound healing, which is generally associated with two processes: chronic wound formation or excessive wound healing [4]. Healing of chronic wounds remains a significant problem for patients and the healthcare system. Therefore, there is a constant search for therapies that can accelerate wound healing.

The wound healing process has many similarities with cancer. They include loss of cell polarity, differentiation, extensive tissue remodeling, and increased cell proliferation. Thus, many molecular and cellular pathways, such as ERK1/2, C-Jun, ROS-associated Ca^2+^, and p53, are common to wound healing and the process of tumorigenesis [5]. Numerous different research methods are currently being explored to accelerate wound healing. Gene therapies seems to be one of the most promising among them. Maternal Embryonic Leucine-zipper Kinase (MELK) and its substrates are known to be involved in the regulation of the cell cycle and cell proliferation. Even though the molecular function is still unknown, MELK is known to be overexpressed in multiple cancers, including melanoma, breast cancer, and renal cell carcinoma [6,7,8]. However, the CRISPRCas9 knockout of this gene in cancer cells had little or no effect on the cell’s proliferation [9]. Moreover, MELK was not found to be a putative cancer target in any of the genetic screens [10,11,12,13]. Therefore, the exact role of MELK in proliferation regulation is still unknown, especially in normal cells. Since wound healing requires increased proliferation of lymphocytes, macrophages, keratinocytes, and fibroblasts, the aim of this study was to evaluate if transient silencing of MELK, using topically applied siRNA, will accelerate wound healing in vivo.

### Innovation

Skin wound healing has been studied for years, but effective therapy that leads to accelerated wound healing remains to be discovered. By evaluating the effect of MELK regulation in wound healing in vivo, we demonstrated that the use of MELK siRNA may be a viable adjunct therapy for accelerated wound healing. A simplified summarizing graphic illustration of the described work is presented in Figure 1.

## 2. Results

In this study, we silenced MELK using topically applied SMARTPool siRNA. The uptake of siRNA by the cells in vivo was confirmed using dye-labeled controls and Azure 400 (Azure Biosystems, Dublin, CA, USA) and is shown in Figure 2.

### 2.1. Wound Size

Significant differences in wound size diameter [mm] upon healing were observed in the MELK siRNA-treated group when compared to the PBS-treated control as measured 3 days after treatment (3.82 ± 0.51 vs. 4.06 ± 0.22, respectively, *p* = 0.0068) and 5 days after treatment (3.66 ± 0.57 vs. 3.94 ± 0.23, respectively, *p* = 0.004). Significant differences in the wound size diameter were also observed on day 7 of the experiment in the MELK siRNA-treated group compared to the non-target siRNA-treated group and the PBS-treated control (2.683 ± 0.82 vs. 3.76 ± 0.23, and vs. 3.53 ± 0.37, respectively, *p* < 0.0001). Eleven, fourteen, and thirty days after the treatment, wounds were fully closed; therefore, no differences were observed between the treated groups. The skin around the wound appeared normal, with no signs of inflammation. The fur growth in the wound area was observed on day 11 of the experiment in all groups. Results are presented in Figure 3.

### 2.2. Immunohistological Staining

Masson’s Trichrome staining performed 3 days after the treatment revealed a significant increase in the presence of total collagen in the wound area of mice treated with MELK siRNA when compared to the non-target siRNA-treated group and PBS-treated control (62 ± 8.22 vs. 31.17 ± 6.15, and vs. 24.80 ± 2.95, respectively, *p* = 0.0001). On day 7, no statistically significant differences were observed. Moreover, no statistically significant differences were observed in the number of mast cells, vessel number, and size of the vessels in the wounds. Results are presented in Figure 4A–D. Representative immunohistological staining images are presented in Appendix A.

### 2.3. Immunofluorescence Staining

The treatment with MELK siRNA resulted in an increased expression, measured using the immunofluorescence score, of Collagen III as compared to PBS-treated control (5.25 ± 1.04 vs. 3.4.17 ± 0.89, respectively, *p* = 0.0347). No statistically significant differences were observed in the cell proliferation rate and apoptosis measured as an immunofluorescence score of Ki67 and TUNEL staining. Moreover, no statistically significant differences in the number of nucleated hematopoietic cells (CD45), myeloid cells (CD11b), helper T cells (CD4), and cytotoxic T cells (CD8a) were observed. Results are presented in Figure 5A–C and Figure 6A–D. Representative immunofluorescence staining images are presented in Appendix A.

## 3. Discussion

The skin, the largest human organ, is constantly exposed to various types of damage, leading to wound formation, which must be constantly healed. Therefore, an efficient wound-healing process is essential for the proper functionality of the body. It is a highly complex process that has been extensively studied and is comprehensively described in the literature [14,15,16,17,18]. Wounds are caused by external factors, such as injuries, cuts, and burns, or internal factors, such as existing pathological conditions such as diabetes. Clinically, depending on the healing time, the wounds are chronic or acute. Even though there is no recognized definition of acute and chronic wounds, it is generally accepted that acute wounds go through a normal healing process and require a relatively short time to heal, while chronic wounds are more challenging and may require up to several months for complete wound closure [19]. In this study, we focused on developing a method that can accelerate the wound-healing process, leading to shorter healing times, regardless of the underlying cause of the wound. Genetic therapies can be considered as being highly innovative among the many strategies that are designed to accelerate wound healing. On the other hand, genetic therapies are regarded as risky due to the possible off-target effects, vector genome integration, and general safety issues. Therefore, in this study, we investigated a vectorless, transient knockdown of the MELK gene, which is known to regulate cell cycle and cell proliferation [20]. This approach allowed us to limit potential adverse effects during the modulation of gene expression. Treatment with MELK siRNA leads to significantly faster wound closing associated with increased collagen expression. There is not much data in the literature regarding MELK’s influence on collagen production, but it was shown by Muller et al. that OTSSP167, a MELK inhibitor, increases collagen deposition by osteoblasts [21]. This data confirms that there is a link between MELK and collagen expression. Furthermore, data indicates that not only is the total collagen concentration increased, but, specifically, collagen type III is. Type III collagen is the first to be synthesized and dominates the early stages of wound healing. It is later replaced by collagen type I, which is the most abundant [22,23]. Finally, MELK knockdown-mediated increased collagen deposition seems beneficial, since collagen plays an essential role in all phases of wound healing, attenuates pro-inflammatory macrophage polarization, and promotes anti-inflammatory macrophage polarization [23,24].

The lack of a targeted vector during potential siRNA therapy may influence off-target cells, for instance infiltrating immune cells [25]. Inadvertently affecting the balance of immune system cells in the wound area may lead to the occurrence of adverse effects, such as a prolonged inflammatory response [26]. In this study, the treatment with MELK siRNA did not affect the number of hematopoietic cells (CD45), myeloid cells (CD11b), helper T cells (CD4), and cytotoxic T cells (CD8a), showing that this therapy does not affect the immune balance of the healing wound [27,28].

As with every gene-expression-modulating therapy, there is an inherited carcinogenicity risk [29]. Even though this risk is relatively small for siRNA-based gene expression modulation due to its transient nature and relatively short half-life, currently, there are no clear regulations defining the carcinogenicity potential of siRNA therapies [29,30]. Moreover, MELK is associated with cancer progression and aggressiveness [31]. Since the balance between apoptosis and proliferation plays a crucial role in tumor formation and growth, we evaluated cell proliferation and apoptosis in the wound. No long-term differences between the study and control groups (up to 30 days after the treatment) were observed, suggesting a negligible carcinogenicity risk.

In conclusion, the results suggest that MELK siRNA treatment leads to accelerated wound healing, with increased collagen deposition apparently being the primary mechanism of action. Moreover, the proposed treatment does not affect the balance of immune cells in the wound area. Finally, no abnormal proliferation or apoptosis, along with no changes in the morphology of the skin, was observed, indicating a low carcinogenicity risk. Even though rodent skin has a similar dermal and epidermal architecture to human skin, murine skin is characterized by a high laxity and mobility of underlying tissue and thus different mechanisms of wound healing; therefore, before this potential MELK-based therapy can be used to accelerate wound healing, more data is needed to evaluate the translational potential for human patients [32,33].

### Key Findings


-siRNA-mediated MELK knockdown leads to accelerated wound healing in mice;-siRNA-mediated MELK knockdown leads to increased collagen deposition in the wound area;-MELK siRNA treatment does not affect the balance of immune cells in the wound area;-MELK siRNA treatment does not affect the long-term apoptosis or proliferation rate of the cells in the wound, indicating a low carcinogenicity risk.


## 4. Material and Methods

The electronic laboratory notebook platform was not used during this study.

### 4.1. siRNA

Accell self-delivering SMARTPool siRNA against mice Melk (accession NM_010790 and XM_006537640), as well as 6-FAM-labeled non-targeting control siRNA and Cy3-labeled anti-cyclophilin B control siRNA (accession NM_011149) were purchased from Horizon Discovery (Cambridge, UK) and used according to the manufacturer’s protocol. The chemical modifications of Accell siRNA were shown to be stable and efficiently delivered to the cells [34]. The optimal concentration of siRNA to be used in in vivo experiments was chosen based on the in vitro study using L929 mice fibroblasts and dye-labeled controls. Briefly, 5000 cells/well were seeded in 96-well culture plates. After 24 h, the culture media was changed, and fluorescent anti-cyclophilin B control siRNA and non-targeting control siRNA were added to the cells in concentrations of 0.5, 1, and 2 µM. After 24 h incubation, the uptake of siRNA was evaluated using a Zeiss Axio Observer (Zeiss, Poland) fluorescent microscope. A concentration of 1 μM was chosen for in vivo experiments.

### 4.2. Animal Experiments

All animal experiments were conducted according to the Declaration of Helsinki as well as Polish regulations and standards for the wellness of laboratory animals. All experiments were accepted by and conducted according to the ethical guidelines of the Local Bioethical Committee (WAW2/050/2019).

The experiments were performed on 16 male, 8-week-old C57BL6 mice that weighed around 25 g each and had unlimited access to feed and water. Mice were obtained from the Center for Experimental Medicine, Medical University of Bialystok. Mice were maintained under conventional conditions of 22.5–23 °C, a relative humidity of 50–70%, and a 12 h day/night cycle.

On the day of the experiment, mice were premedicated with 100 mg/kg of Ketamine and 10 mg/kg of Xylazine, and the fur on the animal’s back was closely clipped over a sufficiently large test area, avoiding mechanical irritation and trauma. The skin was disinfected with an iodine solution. Two full-thickness wounds were generated on both flanks of the animal at equal intervals (four wounds in total) using a 4 mm biopsy punch. A silicone ring was attached around each wound using a tissue adhesive. Based on the in vitro screening test using mice fibroblasts, 1 µM of appropriate siRNA in 20 µL of PBS or 20 µL of PBS was applied to each wound. After 20 min, the wounds were covered with a dressing. Animals were housed separately with unlimited access to feed and water containing paracetamol (3.5 mg/mL for 3 days). After 3 days, the dressing and silicone rings were removed. Wounds were measured in three planes on days 3, 5, 7, 11, 14, and 30 after treatment. To acquire skin samples for further testing, mice were sacrificed by pentobarbital injection and cervical dislocation on days 3, 7, 14, and 30. Each wound with a sufficient margin of healthy tissue was fixed overnight in 10 mL of 4% PFA at 4 °C. Formalin-fixed, paraffin-embedded tissue blocks were created, and 5 µm sections were prepared.

### 4.3. Immunohistological Staining

For histologic analysis, slides were prepared using Masson’s Trichrome Stain, Mast Cell Tryptase antibody, and von Willebrand Factor/Factor VIII complex antibody (Dako) and counterstained with hematoxylin and eosin (H&E) using Dako Omnis automated platform. Slides were scanned at 400× magnification using a Pannoramic 250 Flash II scanner (3DHistech, Budapest, Hungary) and were evaluated using CaseCenter software with QuantCenter extension (3DHistech, Hungary). Up to five hotspots in the wound area per slide were analyzed as described by Weidner et al. [35]. Collagen evaluation was performed using the “Pattern Recognition” tool of the QuantCenter package. Each hotspot field was classified in terms of occupancy by structures automatically recognized as collagen. Mast cells were counted in the hotspot fields by the user. During the evaluation of angiogenesis, each vessel was outlined and sized automatically in CaseCenter. The summed cross-sectional areas of the vessels were related to the size of the hotspot field. The results for each slide were averaged against the indicated hotspot fields.

### 4.4. Immunofluorescence Staining

Immunofluorescence staining was performed as previously reported [36]. Briefly, slides were deparaffinized in xylene and rehydrated by washing in serially diluted ethanol and then distilled H_2_O. Antigen retrieval was performed at 95–100 °C for 20 min using pre-warmed 9.8 mM sodium citrate buffer. Slides were rinsed in distilled H_2_O and PBS, blocked in blocking solution, and stained with primary antibodies overnight in a humidified chamber at 4 °C. Next, the slides were washed in PBS (3 × 5 min), stained with secondary antibody (if necessary) for 2 h at room temperature, washed, and mounted with ProLong Diamond reagent having DAPI. Images were acquired at 200× microscopic magnification using a Nikon A1R confocal microscope. A combinative semiquantitative scoring system was used for fluorescence microscopy results analysis [37]. Immunofluorescence scores were calculated by multiplying intensity scores and extent scores. The percentage of positive cells was divided into five grades (extent scores): 1–5% (1), 6–25% (2), 26–50% (32), 51–75% (4), and 76–100% (5). The intensity of staining was divided into four grades (intensity scores): no signal (0), weak signal (1), moderate signal (2), and strong signal (3). Slides were stained with Ki67 (# 41-5698-82), CD11b (# 53-0112-82), CD4 (# 41-0042-82), CD8a (# 50-0081-82), collagen III (# PA5-34787), and CD45 (# 14-0451-82). Apoptotic cells were evaluated using a Click-iT Plus TUNEL Assay Kit (# C10617) according to the manufacturer’s recommendations. Reagents and antibodies were purchased from ThermoFisher Scientific, Warsaw, Poland.

### 4.5. Statistical Analysis

All results were presented as the mean ± range. The data distribution was evaluated using the Shapiro–Wilk test. Statistical evaluation of the wound size was performed using a two-way ANOVA with Tukey’s multiple comparisons test. Masson’s Trichrome Stain, Mast Cell Tryptase, and von Willebrand Factor/Factor VIII complex staining results were evaluated using the Brown–Forsythe and Welch ANOVA test. Collagen III, CD45, CD11b, CD4, CD8a, TUNEL, and Ki67 staining results were analyzed using the Kruskal–Wallis test. GraphPad Prism software (version 9.4.1; GraphPad Software, Inc., La Jolla, CA, USA) was used for all evaluations. *p* < 0.05 was considered statistically significant.

## Figures and Tables

**Figure 1 ijms-24-01326-f001:**
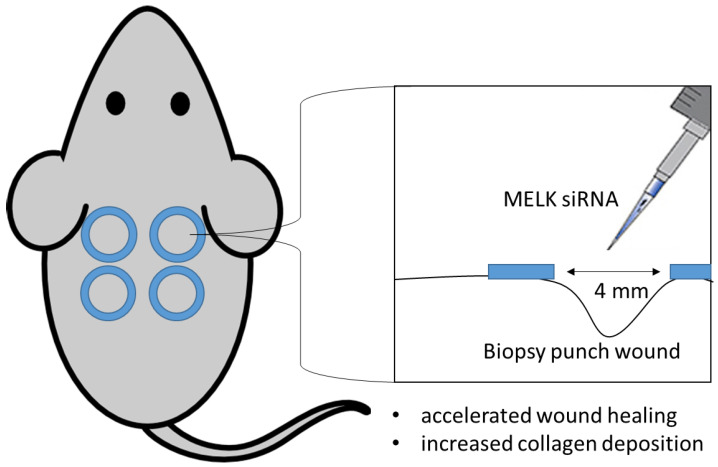
A simplified summarizing graphic illustration of MELK knockdown using MELK siRNA in murine wound healing model.

**Figure 2 ijms-24-01326-f002:**
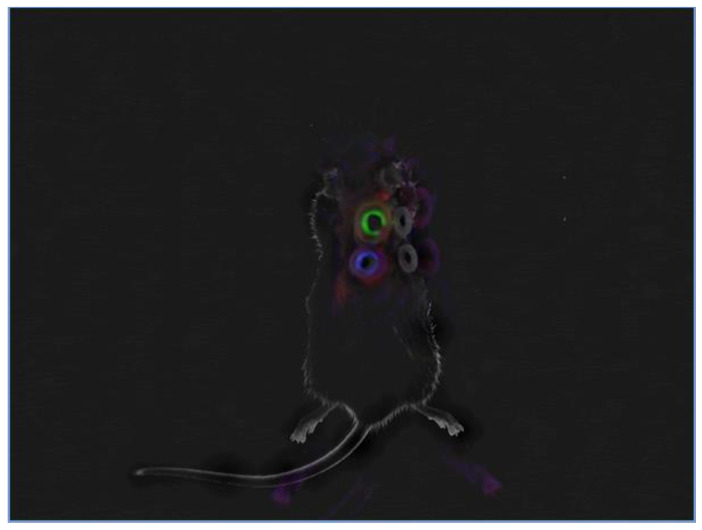
The uptake of siRNA in the wound area. Left side: 6-FAM-labeled non-targeting control siRNA (green), and Cy3-labeled anti-cyclophilin B control siRNA (blue); right side: PBS treatment. A fluorescent signal indicates the uptake of siRNA by the cells in the wound area. No signal is observed in the PBS control group. Analysis was performed using the Azure 400 system.

**Figure 3 ijms-24-01326-f003:**
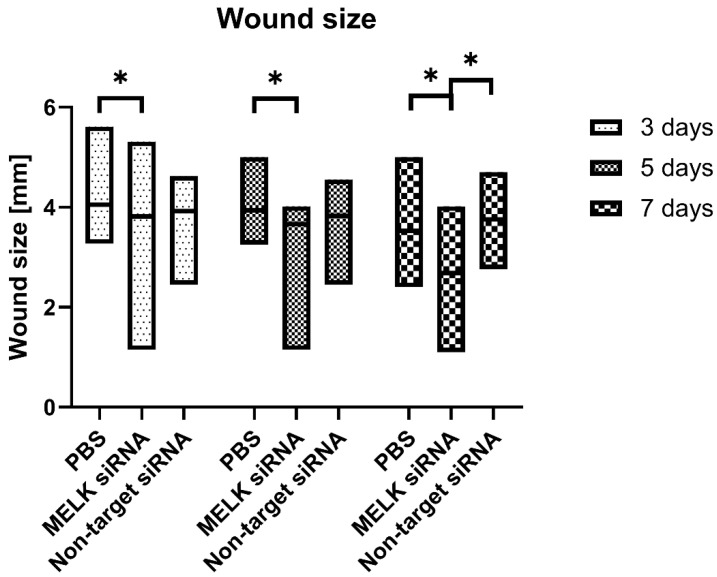
Wound size [mm] after treatment with PBS, MELK siRNA, and non-target siRNA was measured on days 3, 5, and 7 after the start of the experiment. n = 16 for day 3 and n = 12 for days 5 and 7. Results are presented as the mean ± range. *—*p* < 0.05.

**Figure 4 ijms-24-01326-f004:**
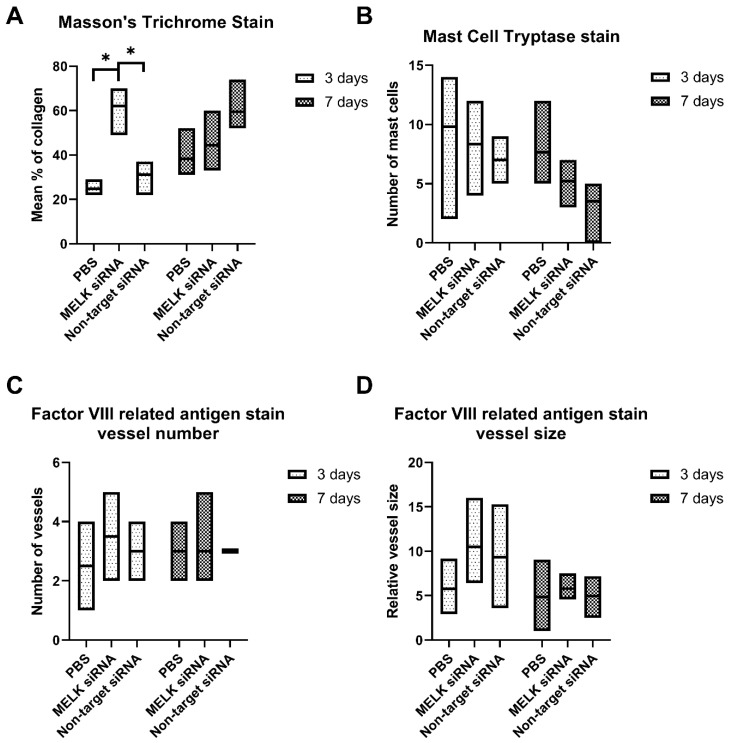
Immunohistochemistry staining results of wounds, 3 and 5 days after treatment with PBS, MELK siRNA, and non-target. (**A**) % of total collagen measured by Masson’s Trichrome Stain. (**B**) the number of mast cells measured by Mast Cell Tryptase stain. (**C**) Number of blood vessels measured by Factor-VIII-related antigen stain. (**D**) Relative vessel size measured by Factor-VIII-related antigen stain. n = 6. Results are presented as the mean ± range. *—*p* < 0.05.

**Figure 5 ijms-24-01326-f005:**
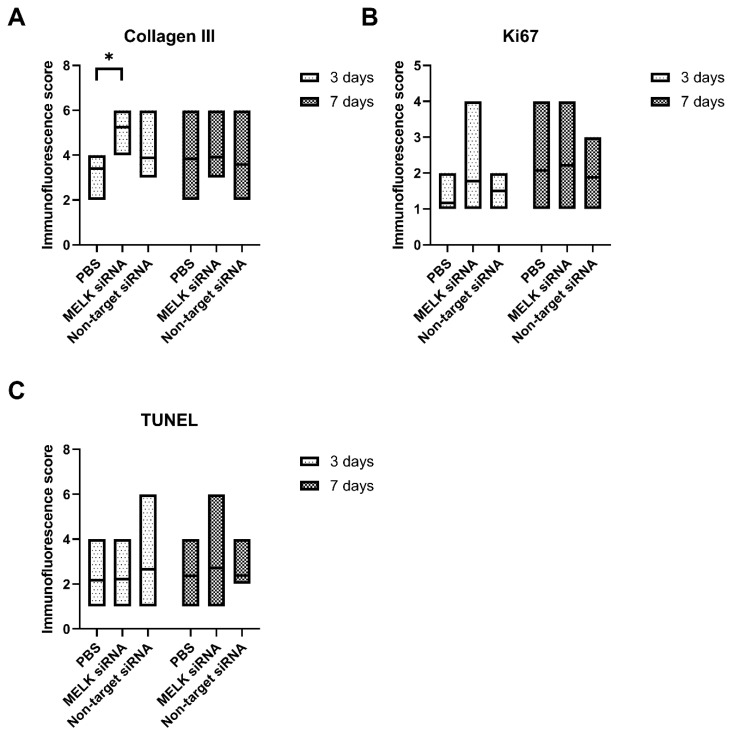
Immunofluorescence staining results of wounds 3 and 7 days after treatment with PBS, MELK siRNA, and non-target. (**A**) Expression of Collagen III. (**B**) Proliferation activity analysis measured as immunofluorescence score of Ki67. (**C**) Apoptosis analysis measured as immunofluorescence score of TUNEL staining. n ≥ 6. Results are presented as the mean ± range. *—*p* < 0.05.

**Figure 6 ijms-24-01326-f006:**
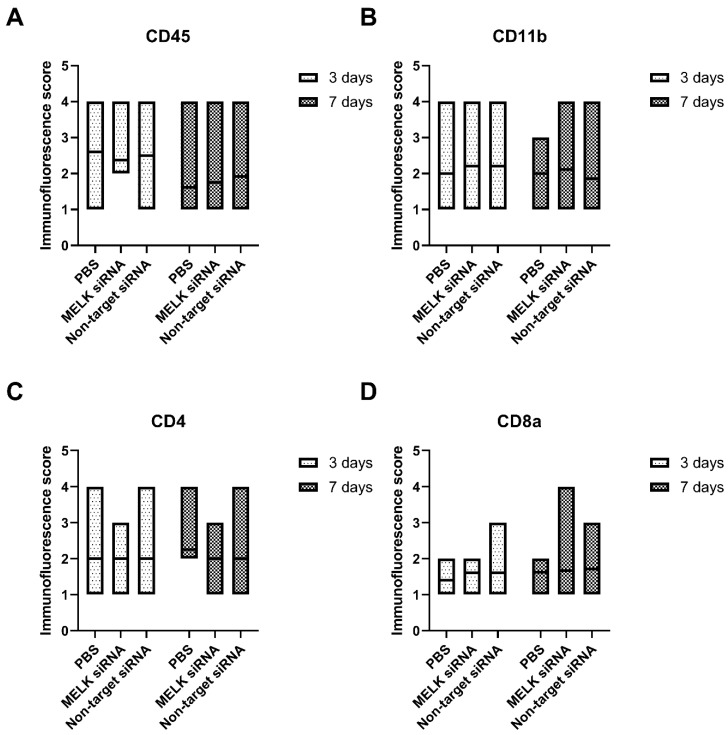
Immunofluorescence staining results of wounds 3 and 7 days after treatment with PBS, MELK siRNA, and non-target. (**A**) Nucleated hematopoietic cell infiltration measured as immunofluorescence score of CD45 expression. (**B**) Myeloid cell infiltration measured as immunofluorescence score of CD11b expression. (**C**) Helper T cells infiltration measured as immunofluorescence score of CD4 expression. (**D**) Cytotoxic T cells infiltration measured as immunofluorescence score of CD8a expression n ≥ 6. Results are presented as the mean ± range.

## Data Availability

The datasets generated during and/or analyzed during the current study are available from the corresponding author upon reasonable request.

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
