# Peer review of "siRNA-Mediated MELK Knockdown Induces Accelerated Wound Healing with Increased Collagen Deposition"

_ijms, 2023, doi:10.3390/ijms24021326_

Round 1

Reviewer 1 Report

Szymanski et al. aimed to evaluate the potential of MELK silencing to accelerate wound healing. They performed an experiment in a murine model of cutaneous wound healing. Szymanski et al. evaluated a variety of markers indicating the progression of wound healing. In my opinion, the experiment was designed well and the methods section is explained in a detailed manner. The introduction and Discussion sections are short but informative. Please find minor comments below to improve the manuscript.

Line 81 – please verify the country based on the company’s headquarter location - in this place and through the article

Line 102 – please provide for how long mice were treated with paracetamol

Line 127 – there is lack of an opening bracket

Figure 3 - authors observed a statistically significant decrease in wound size in the MELK siRNA group in comparison to the PBS group but not to non-target siRNA at days 3 and 5 – please discuss.

Figure 4A – authors observed interesting results regarding collagen depositions in MELK siRNA group. Masson’s Trichrome staining performed 3 days after the treatment revealed a significant increase in the presence of total collagen in the wound area of mice treated with MELK siRNA when compared to the non-target siRNA-treated group and PBS-treated control. However, there is a lack of representative photos of the Masson Trichrome staining for each group. Please provide the pictures.

Figure 5 – Please provide more information in the methodology section about the staining of type III collagen. I know that it was done by the “Pattern Recognition” tool of the QuantCenter package, however, more details would be useful for people not familiar with this System. On what basis it determined the differences between type I and type III collagen?

Figure 6 – in my opinion, letter A in the figure labeling is smaller than B – please verify. This also applies to Figures 4 and 5

Author Response

Response to the Reviews

We want to thank the reviewers for all their valuable comments that allowed us to improve our manuscript. All changes of earlier text are marked in tracking changes mode and are included in the current manuscript text according to all reviewer suggestions. Below, we also include all reviewers' comments with our responses.

Szymanski et al. aimed to evaluate the potential of MELK silencing to accelerate wound healing. They performed an experiment in a murine model of cutaneous wound healing. Szymanski et al. evaluated a variety of markers indicating the progression of wound healing. In my opinion, the experiment was designed well and the methods section is explained in a detailed manner. The introduction and Discussion sections are short but informative. Please find minor comments below to improve the manuscript.

  1. Line 81 – please verify the country based on the company’s headquarter location - in this place and through the article

Answer: We have verified the country of the company throughout the manuscript. We stated the location of the company which we contacted while acquiring equipment/reagents.

  1. Line 102 – please provide for how long mice were treated with paracetamol

Answer: We have introduced this information in the manuscript.

  1. Line 127 – there is lack of an opening bracket

Answer: We revised that.

  1. Figure 3 - authors observed a statistically significant decrease in wound size in the MELK siRNA group in comparison to the PBS group but not to non-target siRNA at days 3 and 5 – please discuss.

Answer: We did not observe significant differences between the non-target siRNA group and MELK siRNA group on days 3 and 5, but the difference was observed after 7 days. MELK siRNA group wound size was significantly smaller than PBS group at day 3 and day 5, while the non-target group was statistically indistinguishable from PBS control at all investigated time points therefore it can be concluded that non-target siRNA did not promote wound healing. As to the reason why there are no statistically significant differences between non-target siRNA group and MELK siRNA group at days 3 and 5 we suspect that the nature of statistical analysis might be the culprit. It is possible that the statistical significance would be more apparent with a higher number of samples. Moreover, as the results indicate, the use of MELK siRNA promotes wound healing through increased collagen deposition and not by direct promotion of cell proliferation; hence the effects may be more pronounced at the later stages of wound healing.

  1. Figure 4A – authors observed interesting results regarding collagen depositions in MELK siRNA group. Masson’s Trichrome staining performed 3 days after the treatment revealed a significant increase in the presence of total collagen in the wound area of mice treated with MELK siRNA when compared to the non-target siRNA-treated group and PBS-treated control. However, there is a lack of representative photos of the Masson Trichrome staining for each group. Please provide the pictures.

Answer: We have prepared Figure S1 containing representative immunohistological images of all evaluated analytes, and Figure S2, containing representative immunofluorescence images of all evaluated analytes. Figures are present in the supplement file.

  1. Figure 5 – Please provide more information in the methodology section about the staining of type III collagen. I know that it was done by the “Pattern Recognition” tool of the QuantCenter package, however, more details would be useful for people not familiar with this System. On what basis it determined the differences between type I and type III collagen?

Answer: Data presented in Figure 5 comes from immunofluorescence staining using specific collagen III (# PA5-34787) antibody. Pattern Recognition tool of the QuantCenter package was used for the evaluation of immunohistological stainings including total collagen using the Masson staining.

  1. Figure 6 – in my opinion, letter A in the figure labeling is smaller than B – please verify. This also applies to Figures 4 and 5

Answer: We have revised that.

Reviewer 2 Report

Mauscript entitled "siRNA mediated MELK knockdown induces accelerated wound healing with increased collagen deposition"  described very innovative approach of topical application of gene therapy for wound healing. Lately several scientists are studying such approach. 

Authors mentioned that at 11, 14 and 30 days after treatment, wound were fully closed therefore no difference were observed between treated groups. Was the same difference observed in control group too?

It is important to find collagen III and collagen I ratio during wound healing process. It is missing in this study. 

How the authors dealt with siRNA degradation, stability, membrane penetration and integration in the cells as they are using topical application without any delivery system.

What is a half life of siRNA? How it will work in more pronounced wound like Diabetic wound and burn wound as penetration into the cells will be difficult?

It would have been appropriate to include known treatment (traditional drug) group as a comparator. 

Manuscript need English revision as there are few spelling mistakes.  

Author Response

Response to the Reviews

We want to thank the reviewers for all their valuable comments that allowed us to improve our manuscript. All changes of earlier text are marked in tracking changes mode and are included in the current manuscript text according to all reviewer suggestions. Below, we also include all reviewers' comments with our responses.

Mauscript entitled "siRNA mediated MELK knockdown induces accelerated wound healing with increased collagen deposition"  described very innovative approach of topical application of gene therapy for wound healing. Lately several scientists are studying such approach. 

  1. Authors mentioned that at 11, 14 and 30 days after treatment, wound were fully closed therefore no difference were observed between treated groups. Was the same difference observed in control group too?

Answer: Yes, there were no differences between the control PBS group, non-target siRNA group and MELK siRNA group. Therefore, the observation of wound size focused on earlier time points, 3, 5 and 7 days after the start of the treatment.

  1. It is important to find collagen III and collagen I ratio during wound healing process. It is missing in this study. 

Answer: We agree that both collagen III and collagen I are important in wound healing. For that reason we evaluated the Collagen III using fluorescence microscopy and total collagen using light microscopy. Knowing that MELK siRNA influences collagen deposition we will include detailed comparison of collagens production, including collagen III to collagen I ratio in future studies.

  1. How the authors dealt with siRNA degradation, stability, membrane penetration and integration in the cells as they are using topical application without any delivery system.

Answer: To ensure siRNA stability, cell uptake of siRNA, and efficient knockdown, we used Accell self-delivering siRNA smart pool. The chemical modifications of siRNA guarantee stability, efficient uptake and knockdown of target gene as proven in a number of publications such as "Use of Self-Delivery siRNAs to Inhibit Gene Expression in an Organotypic Pachyonychia Congenita Model" by Hickerson et al. We introduced an additional description of Aacell siRNA in the M&M section.

  1. What is a half life of siRNA? How it will work in more pronounced wound like Diabetic wound and burn wound as penetration into the cells will be difficult?

Answer: Available data indicates that accell siRNA knockdowns last at least 9 day in rodents as described by Taniguchi et al "Novel use of a chemically modified siRNA for robust and sustainable in vivo gene silencing in the retina". Moreover, passive delivery strategy allows for repeated application of siRNA without the cytotoxic effects related to lentiviral or lipid-based transfections. Therefore, in more pronounced wound like diabetic wound, siRNA can be reapplied with each dressing change. Of course, more research is needed before such therapy can be applied for treatment of diabetic wound in clinical practice. We plan to investigate this area further once we secure funding for the research.

  1. It would have been appropriate to include known treatment (traditional drug) group as a comparator. 

Answer: Presented research was performed as basic studies to explore the concept of MELK manipulation in wound healing. To evaluate the efficiency of treatment, further studies are required, including experiments comparing the proposed treatment to a marketed comparator. As for diabetic wounds, we plan to explore further the area of wound treatment using MELK manipulation.

  1. Manuscript need English revision as there are few spelling mistakes.  

Answer: We revised the entire manuscript.

Reviewer 3 Report

The authors knocked out gene MELK via siRNA from a murine skin wound model and found that wound closure and collagen â…¢ deposition were promoted. The manuscript should be improved before publishing.

Detailed comments/suggestions:

1. Edit errors:

In the line 25: “Cd 45”, in the line 82: “uM” or μM, in the line 91: “22.5-23℃” should have space between number and unit.

Please double check the whole manuscript and revise.

2. In the line 46-47: please give examples that the common pathways between wound healing and tumorigenesis.

3. In the line 125-126: antigen retrieval temperature and time should be given.

4. In the line 151-153: More detailed descriptions for Figure 2 should be given. It should be noted that the difference before and after silencing MELK. How did you qualify the knockout efficiency? Why is the 1 uM concentration of siRNA the best for the experiment?

5. Figure 3: Did the MELK siRNA group and non-target siRNA group have significant differences at day 3 and day 5? As we known, the proliferation stage of the wound healing should be in 3~ 5 days. If there were no differences between the two groups, did it mean that the non-target SiRNA also promote the wound healing?

6. In section 3.2 Immunohistological staining, can you show some staining images, especially for Masson's trichrome staining, because it was the only data that had significant difference?

7. Figure 5: Similarly, can you provide some staining images of collagen III?

Author Response

Response to the Reviews

We want to thank the reviewers for all their valuable comments that allowed us to improve our manuscript. All changes of earlier text are marked in tracking changes mode and are included in the current manuscript text according to all reviewer suggestions. Below, we also include all reviewers' comments with our responses.

The authors knocked out gene MELK via siRNA from a murine skin wound model and found that wound closure and collagen â…¢ deposition were promoted. The manuscript should be improved before publishing.

Detailed comments/suggestions:

  1. Edit errors:

In the line 25: "Cd 45", in the line 82: "uM" or μM, in the line 91: "22.5-23℃" should have space between number and unit. Please double check the whole manuscript and revise.

Answer: We revised the entire manuscript.

  1. In the line 46-47: please give examples that the common pathways between wound healing and tumorigenesis.

Answer: We revised that.

  1. In the line 125-126: antigen retrieval temperature and time should be given.

Answer: We revised that.

  1. In the line 151-153: More detailed descriptions for Figure 2 should be given. It should be noted that the difference before and after silencing MELK. How did you qualify the knockout efficiency? Why is the 1 uM concentration of siRNA the best for the experiment?

Answer: Before studying the effects of 1 uM siRNA in vivo we performed an in vitro pretest to evaluate the optimal concentration of siRNA to be used. Using fluorescently labeled controls and fluorescent microscopy, we chose the optimal concentration based on the signal strength and lack of cytotoxicity. What is more, we confirmed the uptake of siRNA in vivo as presented in Figure 2. Moreover, we used self delivery accell smartpool MELK siRNA which was validated to knockdown at least 75% of MELK. Finally, we amended the description of Figure 2 to be more informative.

  1. Figure 3: Did the MELK siRNA group and non-target siRNA group have significant differences at day 3 and day 5? As we known, the proliferation stage of the wound healing should be in 3~ 5 days. If there were no differences between the two groups, did it mean that the non-target SiRNA also promote the wound healing?

Answer: We did not observe significant differences between the non-target siRNA group and MELK siRNA group on days 3 and 5, but the difference was observed after 7 days. MELK siRNA group wound size was significantly smaller than PBS group at day 3 and day 5, while the non-target group was statistically indistinguishable from PBS control at all investigated time points therefore it can be concluded that non-target siRNA did not promote wound healing. As to the reason why there are no statistically significant differences between non-target siRNA group and MELK siRNA group at days 3 and 5 we suspect that the nature of statistical analysis might be the culprit. It is possible that the statistical significance would be more apparent with a higher number of samples. Moreover, as the results indicate, the use of MELK siRNA promotes wound healing through increased collagen deposition and not by direct promotion of cell proliferation; hence the effects may be more pronounced at the later stages of wound healing.

  1. In section 3.2 Immunohistological staining, can you show some staining images, especially for Masson's trichrome staining, because it was the only data that had significant difference?

Answer: We have prepared Figure S1 in the supplement file containing representative immunohistological images of all evaluated analytes.

  1. Figure 5: Similarly, can you provide some staining images of collagen III?

Answer: We have prepared Figure S2 in the supplement file containing representative immunofluorescence images of all evaluated analytes.

Round 2

Reviewer 3 Report

The authors had addressed the issues that I put forward.